# Genomic and Phenotypic Analysis of Linezolid-Resistant *Staphylococcus epidermidis* in a Tertiary Hospital in Innsbruck, Austria

**DOI:** 10.3390/microorganisms9051023

**Published:** 2021-05-10

**Authors:** Silke Huber, Miriam A. Knoll, Michael Berktold, Reinhard Würzner, Anita Brindlmayer, Viktoria Weber, Andreas E. Posch, Katharina Mrazek, Sarah Lepuschitz, Michael Ante, Stephan Beisken, Dorothea Orth-Höller, Johannes Weinberger

**Affiliations:** 1Institute of Hygiene and Medical Microbiology, Medical University of Innsbruck, 6020 Innsbruck, Austria; silke.huber@i-med.ac.at (S.H.); miriam.knoll@i-med.ac.at (M.A.K.); michael.berktold@i-med.ac.at (M.B.); reinhard.wuerzner@i-med.ac.at (R.W.); 2Center for Biomedical Technology, Department for Biomedical Research, Danube University Krems, 3500 Krems, Austria; anita.brindlmayer@donau-uni.ac.at (A.B.); viktoria.weber@donau-uni.ac.at (V.W.); 3Ares Genetics GmbH, 1030 Vienna, Austria; andreas.posch@ares-genetics.com (A.E.P.); katharina.mrazek@ares-genetics.com (K.M.); sarah.lepuschitz@ares-genetics.com (S.L.); michael.ante@ares-genetics.com (M.A.); stephan.beisken@ares-genetics.com (S.B.); johannes.weinberger@ares-genetics.com (J.W.); 4MB-LAB—Clinical Microbiology Laboratory, 6020 Innsbruck, Austria

**Keywords:** antimicrobial resistance, linezolid, *Staphylococcus epidermidis*, whole genome sequencing, surveillance

## Abstract

Whole genome sequencing is a useful tool to monitor the spread of resistance mechanisms in bacteria. In this retrospective study, we investigated genetic resistance mechanisms, sequence types (ST) and respective phenotypes of linezolid-resistant *Staphylococcus epidermidis* (LRSE, *n* = 129) recovered from a cohort of patients receiving or not receiving linezolid within a tertiary hospital in Innsbruck, Austria. Hereby, the point mutation G2603U in the 23S rRNA (*n* = 91) was the major resistance mechanism followed by the presence of plasmid-derived *cfr* (*n* = 30). The majority of LRSE isolates were ST2 strains, followed by ST5. LRSE isolates expressed a high resistance level to linezolid with a minimal inhibitory concentration of ≥256 mg/L (*n* = 83) in most isolates, particularly in strains carrying the *cfr* gene (*p* < 0.001). Linezolid usage was the most prominent (but not the only) trigger for the development of linezolid resistance. However, administration of linezolid was not associated with a specific resistance mechanism. Restriction of linezolid usage and the monitoring of plasmid-derived *cfr* in LRSE are potential key steps to reduce linezolid resistance and its transmission to more pathogenic Gram-positive bacteria.

## 1. Introduction

*Staphylococcus epidermidis* is a commensal bacterium of humans, prevalent in skin and mucosa, emerging to frequently cause nosocomial and opportunistic infections [1,2]. The clinical relevance of *S. epidermidis* has increased due to the bacteria’s ability to acquire and develop antimicrobial resistance (AMR) to a broad range of antibiotics and its capability of using horizontal gene transfer to disseminate AMR to the more pathogenic *Staphylococcus aureus* [3]. Linezolid, an oxazolidinone, is effective in a variety of Gram-positive strains; it is often the last-resort antibiotic for infections with multi drug resistant (MDR) Gram-positive cocci and was first approved in 2001 in Europe [4]. Indications for the use of linezolid include mostly MDR cocci associated skin and soft tissue infections as well as pneumonia and bloodstream infection [5]. Linezolid acts as a protein synthesis inhibitor that prevents the formation of the initiation complex by binding to the peptidyl transferase center (PTC) of the 23S rRNA subunit [6]. In general, linezolid shows a rather high susceptibility rate of over 99% to Gram-positive pathogens as observed by both surveillance programs ZAAPS (Zyvox^®^ Annual Appraisal of Potency and Spectrum) [7] and LEADER (Linezolid Experience and Accurate Determination of Resistance) [8]. However, case reports of acquired resistance to linezolid in clinical staphylococcal isolates with different resistance mechanisms were published shortly after its implementation in clinical routine [9]. Nowadays, linezolid-resistant *S. epidermidis* (LRSE) exhibits three main resistance mechanisms: (I) plasmid-derived *cfr* acquisition, encoding a ribosomal methyltransferase; (II) mutations in the ribosomal proteins L3, L4, and L22; and (III) mutations in specific 23S rRNA binding sites for linezolid (e.g., G2576U, G2447U, U2504A, C2434U, and G2631U after *E. coli* numbering) [7,8,10,11,12,13,14]. Besides point mutations and genes effecting the binding site of the target PTC, efflux pumps are suggested to be involved in resistance mechanisms of bacteria against linezolid [15]. Transferable resistance genes in LRSE resulting in a colonization or an infection are increasingly reported in animals and humans [16,17,18,19,20,21]. Thus, LRSE are an emerging global public health problem. At our institution, an increase of LRSE patient isolates has been observed in recent years with a peak of 84 isolates in 2018, despite reduced consumption of linezolid compared to 2012 and 2013 (9315 g in 2018 vs. 10,488 g and 10,032 g in 2012 and 2013, respectively).

Therefore, we conducted a retrospective study investigating linezolid resistance mechanisms in clinical *S. epidermidis* isolates of our tertiary hospital in Innsbruck, Austria, by whole genome sequencing (WGS). Furthermore, we analyzed potential pathogen and host risk factors for the development of linezolid resistance in *S. epidermidis*.

## 2. Materials and Methods

### 2.1. Clinical Isolates, Identification, and Antimicrobial Susceptibility Testing

At the Institute of Hygiene and Medical Microbiology, Medical University of Innsbruck, routinely isolated coagulase-negative Staphylococci (CoNS) from clinical samples were cryoconserved upon detection of linezolid resistance. However, it is noteworthy that no routine screening for linezolid resistance was performed; instead, merely CoNS from sterile body sites or suspected involvement in infections were tested against linezolid. In total, 136 LRSE isolates (one isolate per patient only) selected from a larger collection of routine clinical samples (*n* = 347) collected at our institution between 2011 and 2019 were included in this study. Strains were primarily selected based on patient records indicating no prior administration of linezolid before isolation of a LRSE (*n* = 48), and the remaining isolates were randomly selected for further analysis (*n* = 88). Isolates of linezolid-susceptible *S. epidermidis* (LSSE; *n* = 18) represented the control group for resistance mechanisms analysis. Species identification was performed by matrix-assisted laser desorption/ionization time of flight mass spectrometry (MALDI-TOF MS, Bruker Daltonik, Bremen, Germany) using the reference Biotyper library v4.1 (Bruker Daltonik, Bremen, Germany). Species were confirmed by whole genome analysis (see Section 2.2). Antimicrobial susceptibility testing (AST) was performed by disk diffusion according to the current EUCAST guidelines at the year of isolation [22]. To assess the minimal inhibitory concentration (MIC) for linezolid, a commercially available Etest^®^ (bioMérieux, Marcy l’Etoile, France) was performed. Strains were classified as susceptible (S) and resistant (R) according to the current EUCAST breakpoints [23]. All isolates were appropriately stored in skim milk at −80 °C and cultured on Columbia blood agar for shipment to Ares Genetics for further analysis.

### 2.2. DNA Extraction, Library Preparation, and Whole Genome Sequencing of Isolates

One-hundred fifty-four culture plates with bacterial isolates were sent from the Medical University of Innsbruck to the Ares Genetics laboratory. DNA extraction, library preparation, and WGS were conducted as reported previously [24]. In brief, single colonies were picked from the plates and resuspended in lysis buffer. DNA extraction was done on a QIAsymphony (QIAGEN, Hilden, Germany) with custom settings. A260/280 values as well as DNA concentration were measured. All samples passed the minimum requirement of 2 ng/µL DNA concentration needed for library preparation. 

WGS libraries for 2 × 150 bp Illumina sequencing were prepared using QIAseq FX DNA Library Kit (QIAGEN). Quality control (QC) for NGS libraries included fragment size analysis and fluorometric measurement of concentrations, as well as qPCR quantification for the final pool before sequencing. A non-template control sample was added to the library preparation to detect potential contaminations. Sequencing was done on an Illumina Nextseq platform (Illumina Inc., San Diego, CA, USA). 

### 2.3. NGS Data Processing

QC and de novo genome assembly of raw data were performed for all 154 isolates as described previously [24]. Seven assemblies failed internal QC due to gene duplication rates exceeding 60% (5 cases) and low-quality assembly with N50 < 5000 (2 cases). All except one isolate were confirmed as *S. epidermidis* with Kraken2 [25]. The remaining isolate was identified as *S. haemolyticus,* and thus, excluded. Assemblies of the remaining 129 LRSE and 17 LSSE isolates were used for resistance analysis.

### 2.4. Detection of Linezolid Resistance-Associated Genes and Mutations

Genes and point mutations associated with linezolid resistance were extracted from ARESdb [26] and by literature search. *cfr* (AM408573, AJ879565), *optrA* (ALMH01000001, ALZI01000051), *poxtA* (MF095097), Lsa(A) (AAO43110), Vga(B) (AAB95639), and efflux pumps lmrS (F6MF49), AcrA (P0AE06), EmrD-3 (A0A0H3AF79), and EmrE (CAA77936) were identified via sequence alignment [24]. Known point mutations in 23S rRNA, L3, and L4 (Appendix A) were manually inspected. Specifically, for 23S rRNA, the assemblies’ PTC of 23S rRNA were identified via multiple sequence alignment against 5 copies of 23S in *S. epidermidis* as annotated in ARB SILVA (ATCC 12228). PTC nucleotide sequences were extracted using primer sequences for domain V [27] and multiple sequence alignments were generated using ClustalW [28].

Novel point mutations in 23S rRNA were identified via variant calling. Reads were mapped to 5 copies of 23S in *S. epidermidis* as annotated in ARB SILVA (ATCC 12228) using bwa mem (v0.7.17) [29]. The alignments were de-duplicated with the Picard tools (v2.2.0) [30] and variants were called with bcftools (v1.9) [31].

Detected resistance markers were converted to a presence/absence matrix for statistical analysis. Subsequently, for intuition of the utility of detected genes and mutations in linezolid resistance prediction, a simple decision tree classifier of depth 3 was built with 5-fold stratified cross-validation.

### 2.5. Phylogenetic Computations/Multilocus Sequence Typing

Multilocus sequence types were determined with mlst 2.11 [32] using PubMLST scheme “sepidermidis”. The phylogenetic tree was calculated using Mash distances with Mashtree 1.2.0 [33]. Assembly contigs were stitched together with a spacer of 160 Ns prior to processing.

### 2.6. Data Collection

Retrospectively, patient data of hosts of respective resistant isolates including date of sampling, gender, age, weight, ward, duration of hospitalization in relevant ward, source of isolates, laboratory parameters (aspartate transaminase (ASAT; U/L), alanine transaminase (ALAT; U/L), gamma-glutamyltransferase (γGT; U/L), C-reactive protein (CRP; mg/dL), procalcitonin (PCT; µg/L)), and use of antimicrobial treatment and mortality were collected (Appendix A). To assess prior treatment with linezolid, all discharge letters from within one year before isolation of LRSE were screened for mentions of “Linezolid” and “Zyvoxid” (trade name in Austria). Furthermore, medication lists as well as relevant paragraphs mentioning antibiotic treatment were evaluated. As a control group for epidemiological data, a subset of septic patients (*n* = 133) with a confirmed bloodstream infection and known status of linezolid use between 2013 and 2019 was randomly chosen from a previous study conducted at our institution, investigating parameters that interfere with a specific polymerase chain reaction (PCR)-based diagnostic tool for bloodstream infections [34]. 

### 2.7. Statistical Analysis

Results of descriptive statistics are shown as mean ± standard deviation. Continuous parameters were tested for normal distribution by Shapiro–Wilk test. Isolates were considered independently sampled and independence between categorical MICs, high (≥48 mg/L)/low-level (<48 mg/L) resistance, and sampling year assessed by Fisher’s exact test or Chi-square test. Mean differences were compared using the Mann–Whitney test statistic. Linezolid use analysis was performed with a reduced data set including only isolates with known linezolid use status (*n* = 108). Novel mutations in the 23S rRNA were investigated for association to linezolid resistance by ordinary least squares multivariate regression. MIC values were log2-transformed before models were built on known and novel resistance markers with and without pairwise interaction terms.

## 3. Results

### 3.1. Patients Characteristics

Among the 129 LRSE positive patients, 98 were male (75.97%) and 31 (24.03%) female patients with a mean age of 59.24 ± 14.57 years. No patient was younger than 18 years. The mean weight was 77.83 ± 18.31 kg. Patients were admitted to seven different departments within our tertiary hospital, most of them coming from internal medicine (*n* = 60). Forty-four clinical samples were collected from patients at an intensive care unit (ICU). The average duration of hospitalization in the relevant ward was 48.20 ± 37.61 days, and 19 patients (14.73%) died within one month after pathogen isolation. LRSE-isolates were mainly recovered from catheter tips (*n* = 77) and blood cultures (*n* = 30). Among 108 cases with a known status of linezolid consumption, 47 patients did not receive linezolid, whereas 61 patients received linezolid. Ninety-seven patients were treated with two or more antimicrobial substances in the year prior to the bacterial isolation, and 29 patients received none or only one antibiotic substance. For the three remaining patients, it was unclear whether they received antimicrobial drugs or not. It is noteworthy that isolates were selected for sequencing based on patient records regarding prior linezolid administration. In the original strain collection of LRSE from which these isolates were selected, 222 patient records stated prior linezolid administration, 48 recorded no prior linezolid administration, and 77 were inconclusive.

Comparison of the study population and the septic control group revealed a significantly lower number of female patients in the LRSE group (*n*_LRSE_ = 31, *n*_spetic_ = 48; *p* < 0.05), but no differences in age distribution or other clinical parameters. 

### 3.2. Mechanisms of Resistance to Linezolid Mainly Based on Mutations of the 23S rRNA in ST2 Isolates

In 117 out of 129 resistant isolates, at least one known AMR marker for linezolid resistance was detected. Within the linezolid susceptible control group, no known marker was present. The 23S rRNA mutation G2603U showed the highest prevalence with 70.54% among genotypically resistant isolates, followed by the ribosomal protein L3 mutation H146Q and the *cfr* gene (both 23.26%). The efflux pump EmrE was present in only one isolate. No additional AMR markers were found. In 35 isolates, two AMR markers for linezolid were identified. Interestingly, *cfr* and EmrE were only present in isolates with co-resistances to point mutations in the central loop of domain V of the 23S rRNA (G2603U) or in the ribosomal protein L3 (H146Q) (Table 1).

All 30 *cfr* positive isolates carried a *cfr* marker sequence. Alignments indicated that they descended from the *Staphylococcus warneri* pSCFS6 plasmid with Tn*558* transposon. In addition, in 14 isolates, a *cfr* marker sequence retrieved from *Staphylococcus aureus* partial pSCFS3 plasmid was present. These results propose a plasmid-derived acquisition of linezolid resistance. Interestingly, one third of the isolates (*n* = 35) had more than one independent resistance mechanism present. Furthermore, none of the isolates harbored the *optrA* or *poxtA* gene. However, for 12 isolates, no investigated resistance mechanisms could be identified after the first assembling. For these clinical isolates, ARB-SILVA 23S rRNA reference sequences were used to detect further single nucleotide polymorphisms (SNPs). Hereby, all 12 isolates were found to harbor 83 different SNPs/variants of common 23S rRNA mutations, of which 13 were detected in more than one isolate (Appendix A). Thus, including the variant calling, all resistant isolates exhibited at least one (putative) AMR marker for linezolid. 

For intuition, known markers L3 H146Q, *cfr*, EmrE, and 23S rRNA G2603U were used with and without all novel variants, occurring more than once, in R/S modeling. The decision tree achieved an accuracy of 97.3% (sensitivity = 98.5%, specificity = 88.2%) with the entire set and an accuracy of 91.8% (sensitivity = 90.7%, specificity = 100.0%) without the novel, putative markers. 

Multilocus sequence typing revealed the dominant presence of sequence type (ST) 2 (*n*_LRSE_ = 95, *n*_LSSE_ = 5) within the study population, followed by ST5 (*n*_LRSE_ = 31, *n*_LSSE_ = 6). Additionally, the ST23 (*n*_LRSE_ = 1), ST87 (*n*_LSSE_ = 2), ST190 (*n*_LSSE_ = 1), ST23 (*n*_LSSE_ = 1), and ST32 (*n*_LSSE_ = 1) strains and three novel strains (STnovel) were identified (Figure 1). Of note is that 28 out of 30 *cfr*-carrying isolates were ST2 strains. The remaining two were classified as ST5. Interestingly, the 12 isolates with prior unknown mutation mechanisms were found to be classified in five different ST strains, namely ST2 (*n* = 4), ST23 (*n* = 1), ST5 (*n* = 5), and STnovel (*n* = 2).

### 3.3. Novel Variants in the 23S rRNA Linked to Linezolid Resistance

In line with literature evidence, linear regression identified L3 H146Q and *cfr* as significantly associated with linezolid resistance (*p* < 0.01). Novel variants in the 23S rRNA gene with positive coefficients were extracted at *p* < 0.05 and include A131G, G262U, G1256U, G1273A, and A2395AG. Pairwise interactions did not contribute to elevated MICs. Of the five novel variants, only the insertion of A2395AG maps into Domain V occurred (Appendix A).

### 3.4. Phenotypic Linezolid Resistance Differs by Mechanism

Eighty-three out of 129 clinical isolates showed the highest minimal inhibitory concentration (MIC) for linezolid of ≥256 mg/L. Overall, 73.64% (*n* = 95) of the isolates were classified as high-level linezolid-resistant showing a MIC of ≥48 mg/L (Table 2).

Besides linezolid, all isolates except for one were methicillin-resistant. Additionally, 85.3% (*n* = 110) of the isolates were resistant to fluoroquinolones, whereas only one isolate (0.8%) was tigecycline-resistant, and all were susceptible to vancomycin. Overall, the proportion of high-level linezolid-resistant strains appeared to decrease over the years (Figure 2a). Interestingly, the level of linezolid resistance (high/low) could be associated with the year of sampling (*p* < 0.0001). In contrast, no correlation between the resistance level (high/low) to linezolid with the occurrence of more than one resistance mechanism was found. In addition, no correlation with the level of phenotypical linezolid resistance and the administration of linezolid in the past year was present. However, the presence of the *cfr* gene in isolates phenotypically showed significantly (*p* < 0.0005) higher MICs for linezolid than non-*cfr*-carrying isolates (Figure 2b). High-level linezolid resistance was present in 71.6% (*n* = 68) of ST2, 80.7% (*n* = 25) of ST5, 100% (*n* = 1) of ST23, and 33.3% of STnovel (*n* = 3) strains, showing no significant association between resistance level and ST.

### 3.5. Type of Resistance Mechanism or ST Was Not Dependent on Linezolid Use

Around half (53.8%) of the patients with a *cfr* positive LRSE-infection received linezolid within one year prior to the isolation of the causative pathogen (*n* = 14). The same ratio was found within non-*cfr*-carrying isolates, where 47 out of 82 patients received linezolid (57.3%), resulting in no significant difference between the two groups. Hereby, 2 out of 3 STnovel, 46 out of 83 ST2, and 13 out of 15 ST5 isolates could be associated with the administration of linezolid. However, we found no significant correlation between strain subtypes and the use of linezolid.

## 4. Discussion

In recent years, the surveillance of antimicrobial resistances by WGS has become a useful and WHO-recommended tool to monitor the spread and dynamics of resistance mechanisms in bacteria [35]. Here, we investigated genetic resistance mechanisms and respective phenotypes of 129 LRSE recovered from a cohort of patients receiving or not receiving linezolid within a tertiary hospital in Innsbruck, Austria. Epidemiologically, the gender distribution within our study population was significantly different compared to the septic control group, concluding that the high frequency of LRSE in males is specific and not only a reflection of the general hospitalization rate of elderly patients. 

Isolated LRSE showed at least one putative AMR marker recovery and variant calling. Of note, the 23S rRNA gene can occur several times in a bacterial genome; short-read assemblies may not accurately reflect all gene copies and variants present. In ARB SILVA, five copies of the 23S rRNA gene in *S. epidermidis* (ATCC 12228) can be found. To our knowledge, this study investigated the largest number of LRSE isolates regarding their AMR markers against linezolid (*n* = 129). WGS identified the known point mutation G2603U (*S. aureus* numbering) in the 23S rRNA gene (*n* = 91) as the most common resistance mechanism, followed by the presence of the *cfr* gene (*n* = 30). Susceptible isolates (*n* = 17) did not harbor any AMR markers. As the point mutation G2603U in the 23S rRNA after *S. aureus* numbering is equivalent to the well described G2576U point mutation after *E. coli* numbering [36], our study confirms several reports presenting this specific point mutation to be the most prominent AMR marker for linezolid resistance in Gram-positive bacteria [10,11,12,13,37,38,39,40]. Remarkably, the G2576U mutation was the only known point mutation in the 23S rRNA region within our study population. Regarding the presence of genes affecting the susceptibility of bacteria to linezolid, only *cfr* combined with a point mutation, but no *optrA* or *poxtA* was identified in 129 LRSE isolates. In line with our findings, a study conducted in the United States analyzed 31 LRSE isolated from blood cultures of cancer patients by WGS and linked the presence of *cfr* and mutations in the 23S rRNA (C2576U), L3, and L4 in all isolates to linezolid resistance [41].

We further briefly explored the set of markers and their association with resistance by in silico modeling. For intuition, known markers L3 H146Q, *cfr*, EmrE, and 23S rRNA G2603U were used with and without all novel variants, occurring more than once, in R/S modeling. The decision tree achieved an accuracy of 97.26% with the entire set and an accuracy of 91.78% without the novel, putative markers. Factoring in the limited sample size and imbalanced distribution of LRSE and LSSE isolates, the models confirm the strong link between known markers and linezolid resistance in agreement with previous findings. Regression analysis identified a total of five, to our knowledge, novel variants associated with linezolid resistance in *S. epidermidis* including A131G, G262U, G1256U, G1273A, and A2395AG.

Multilocus sequence typing identified ST2 as dominant strains, followed by ST5, whereas LSSE was diverse. In the literature, clonal occurrence of both ST2 and ST5 are described. Bouiller and co-workers reported a clonal spread of LRSE within an intensive care unit in Germany [42]. However, both dominant multilocus ST of our LRSE strains are frequently associated with linezolid resistance, proposing the ability of these strains to acquire and/or transmit resistance mechanisms.

Amongst several studies, the administration of linezolid has been identified as an independent risk factor for colonization, infection, and isolation with LRSE [42,43,44,45]. Selective pressure promotes the survival of resistant strains, and an overgrowth of these facultative pathogenic bacteria can lead to infections in critically ill or immunocompromised patients. In line with the data, the majority of patients with LRSE at our hospital had received linezolid in the year prior to LRSE isolation (64%; *n* = 222 out of a total of *n* = 347). However, linezolid resistance without prior administration of linezolid during hospitalization had been reported in several other studies where up to 55% of patients with LRSE had no history of linezolid use [44,46,47,48]. Correspondingly, in our patient population, 47 out of 347 individuals had not received linezolid. Noticeably, the G2576U mutation was demonstrated to have a negative effect on the biological fitness of *S. aureus*, and a restoration of the wild-type sequence was observed after cessation of exposure to linezolid [49]. In contrast, *cfr* was shown to induce only low fitness cost in staphylococci, which suggests an easier maintenance of the *cfr* gene compared to G2576U mutations [50]. Together with its frequent location on mobile genetic elements, this provides a more favorable frame for the spread of *cfr* amongst isolates not exposed to selective pressure by linezolid administration [4,51]. Our hypothesis was, thus, that the genetic profiles of strains isolated from patients without prior linezolid administration would differ from those where the selective pressure was apparent. However, in contrast to our hypothesis, no difference was found between the genetic profiles of strains from patients without versus with linezolid application in the year prior to isolation. This indicates a probable transmission of LRSE from one patient to others. Furthermore, despite decreasing administration of linezolid at our institution, total numbers of LRSE isolates increased. These results support the assumption that treatment with linezolid is not the only factor contributing to the rise of linezolid-resistant Gram-positive cocci [42,45]. While the administration of linezolid stays the driving force for selection of resistant strains, our data support the significance of infection control measures to prevent further spread of Gram-positive cocci resistant to linezolid.

While, in our strains, the combination of more than one resistance mechanism per se was not significantly associated with the expression of a high-level resistance phenotype, we demonstrated that the level of resistance was significantly higher when *cfr* was part of the combination. In our strains, *cfr* occurred only in combination with other resistance mechanisms such as L3 H146Q or 23S rRNA G2603U, and was in all cases plasmid-derived. It was experimentally shown that *cfr* alone renders staphylococci low-level linezolid resistance, increasing the MIC to a range of 8 to 32 mg/L, while a combination with other mechanisms elevated the MIC further [50]. The co-occurrence of *cfr* with any other linezolid resistance mechanism was first demonstrated in 2010 [52] and has since been associated with high level resistance when combined with other mechanisms, such as mutations in domain V and in the ribosomal proteins L3 and L4 [14,41,53]. The presence of *cfr* is of special concern as its common location on a plasmid enables the gene to be horizontally transferred and comes with a low fitness cost for the organism [54,55]. While the clinical impact and pathogenicity of LRSE varies greatly, the transfer of linezolid resistance genes such as *cfr* to more pathogenic Gram-positive cocci may limit therapeutic options and deteriorate patient outcome upon infection. Taken together, in our strains, *cfr*-carrying isolates had higher MIC values than isolates without *cfr*, supporting the observation that a combination of *cfr* with other mechanisms leads to the expression of a phenotype with high-level resistance towards linezolid, which is of special relevance, as this resistance gene may be transferred to other Gram-positive cocci.

## 5. Conclusions

In our collection of LRSE strains, a specific point mutation of the 23S rRNA (G2603U) was the most common resistance mechanism followed by the presence of the *cfr* gene. The *cfr* gene occurred only in combination with other resistance mechanisms, and *cfr*-harboring isolates demonstrated higher phenotypic resistance levels compared to other *cfr*-negative strains. Our LRSE strain collection showed a rather clonal population structure with ST2 and ST5 being the most common sequence types. Linezolid administration was identified as the most prominent trigger for the development of linezolid resistance. However, other so far unknown sources exist. Thus, further studies are needed to reveal these triggers. Finally, restriction of linezolid usage and the monitoring of plasmid-derived *cfr* are potential key steps to reduce linezolid resistance and its transmission to more pathogenic Gram-positive bacteria. 

## Figures and Tables

**Figure 1 microorganisms-09-01023-f001:**
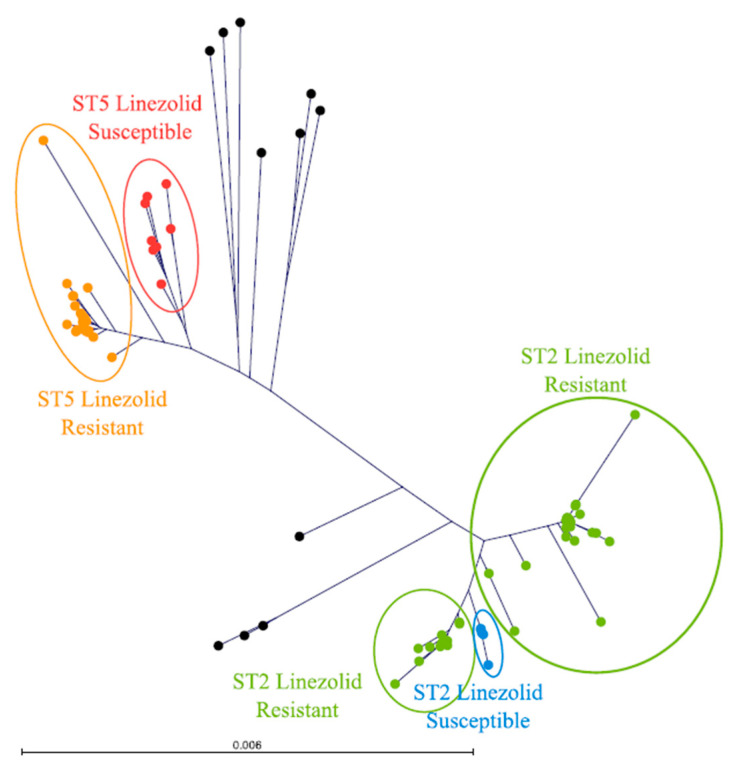
Phylogenetic tree of linezolid-resistant *S. epidermidis* (LRSE) and linezolid-susceptible *S. epidermidis* (LSSE) isolates. Coloring of the tree corresponds to manual partitions for the two dominant sequence types (ST) ST2 and ST5 as well as associated linezolid resistance. Leaves colored in black correspond to ST2, ST5, ST23, ST32, ST87, ST190, and STnovel.

**Figure 2 microorganisms-09-01023-f002:**
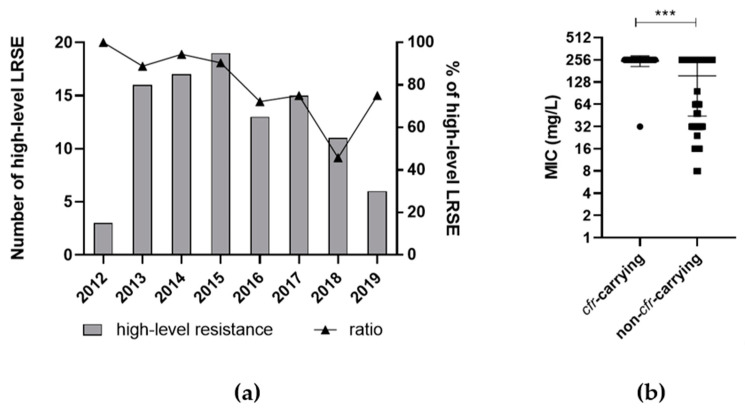
Phenotypically high linezolid resistance levels and minimal inhibitory concentrations (MIC). (**a**) The total amount of high-level linezolid-resistant *S. epidermidis* (LRSE) as well as the percentage of high-level LRSE decreased over the years. (**b**) Presence of *cfr* expressed phenotypically higher MIC levels. MIC break points of *cfr*-carrying and non-*cfr*-carrying isolates were compared by Fisher’s exact test (*** *p* < 0.001).

**Table 1 microorganisms-09-01023-t001:** Emergence of linezolid resistance mechanisms within investigated linezolid-resistant *Staphylococcus epidermidis* (LRSE) isolates. Percentages are given in brackets.

Isolate Count (*n*) per Mutation				
23S G2603U	L3 H146Q	*cfr*	EmrE	*n* (%)
x				58 (45.0)
	x			25 (18.6)
x		x		28 (21.7)
x	x			4 (3.1)
x			x	1 (0.8)
	x	x		2 (1.6)
				12 (9.3) *
91 (70.5)	30 (23.3)	30 (23.3)	1 (0.8)	129

* ARB-SILVA 23S rRNA reference sequences used to detect further SNPs.

**Table 2 microorganisms-09-01023-t002:** Minimal inhibitory concentration (MIC) of linezolid-resistant *Staphylococcus epidermidis* (LRSE) isolates. Percentages are given in brackets.

Level of Linezolid Resistance	MIC (mg/L)	Number of LRSE Isolates
high	≥256	83 (64.3)
	96	1 (0.8)
	64	8 (6.2)
	48	3 (2.3)
low	32	21 (16.3)
	24	1 (0.8)
	16	9 (7.0)
	8	3 (2.3)
Total		129 (100.0)

## Data Availability

*S. epidermidis* sequencing data have been deposited in the National Center for Biotechnology Information (NCBI) under Bioproject ID PRJNA720857. Additional data provided upon request to the corresponding author.

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
