# Peer review of "Genomic and Phenotypic Analysis of Linezolid-Resistant Staphylococcus epidermidis in a Tertiary Hospital in Innsbruck, Austria"

_microorganisms, 2021, doi:10.3390/microorganisms9051023_

Round 1

Reviewer 1 Report

Please add if the data are tested for normal distribution (eg. with Shapiro-Wilk test or similar).

Reviewer 2 Report

The manuscript by Huber and colleagues reports on the whole genome sequence analysis of a large collection of Staph. epidermidis isolates to understand the basis of linezolid resistance in this taxon. The data are well presented (although a couple of clarifications are requested) and informative. The comments below are primarily language suggestions.

  1. L16. By specifically stating “Gram positive”, this could be interpreted to mean that WGS analysis is not appropriate for Gram negatives. Perhaps just use “in bacteria”?
  2. L17 sequence types
  3. L19 and throughout. These mutations are in the “23S rRNA gene” rather than having a “T” in 23S rRNA. Also, in lines 306 and 353 the Svedberg abbreviation is not capitalized (please check throughout).
  4. L34 of SE has increased
  5. L79. Please include the MALDI platform (Bruker?)
  6. L84. Do the authors mean “appropriately” rather than “adequately”?
  7. L92 Introduce “Qiagen” here. Also, depending on journal policy, locations of companies may be required on first use (see also L100 where Illumina location may need to be provided).
  8. L67 insert the abbreviation “MUI” here as the abbreviation is not introduced before first use L89
  9. L103 were performed
  10. L136 Collection (to match capitalization of other sections). This may also be needed for ‘dependent’ L269.
  11. L157 was performed with a reduced (two changes)
  12. L179 48 recorded no
  13. Table 1 cfr should either be capitalized for protein) or italicized (for gene)
  14. L201 and L203. The authors should mention why these annotations are in quotation marks.
  15. L201-L203. It is not clear to what extent a sequence or contig matches a plasmid. Are the authors indicating that a whole plasmid contig matched a plasmid? Or if cfr was found on a contig with a specific rep gene, was it called as a plasmid? These few sentences need to be expanded upon/clarified, so that the reader can evaluate how the contigs or cfr sequences are assigned to plasmids.
  16. L202 558 should be italicized
  17. L222 two were classified
  18. L234 Novel variants in the 23S rRNA gene
  19. L246 fluoroquinolone L247 tigercycine
  20. L250 In contrast
  21. L271 prior to
  22. L273 resulting in no significant
  23. L279. Please provide a reference for the WHO recommendation
  24. L278 WGS has become a
  25. L287 Of note, the 23S rRNA gene
  26. L289 five copies of the 23S rRNA gene L292 23S rRNA gene
  27. L353 mechanisms such as L363 such as cfr
  28. L396 Bioproject
